# Detecting the Dynamics of Urban Growth in Africa Using DMSP/OLS Nighttime Light Data

Shengnan Jiang [1], Guoen Wei [1], Zhenke Zhang [1,2,*], Yue Wang [1,2], Minghui Xu [1], Qing Wang [1,2], Priyanko Das [1] and Binglin Liu [1,2]

1 School of Geographic & Oceanographic Sciences, Nanjing University, Nanjing 210023, China; DZ1627002@smail.nju.edu.cn (S.J.); dg1927034@smail.nju.edu.cn (G.W.); DG20270035@smail.nju.edu.cn (Y.W.); DZ1727008@smail.nju.edu.cn (M.X.); DG1627044@smail.nju.edu.cn (Q.W.); DG1927501@smail.nju.edu.cn (P.D.); DG1827018@smail.nju.edu.cn (B.L.)
2 The Collaborative Innovation Center of South China Sea Studies, Nanjing University, Nanjing 210093, China
* Correspondence: ZHANGZK@nju.edu.cn; Tel.: +86-25-8968-6694

**Abstract:** Africa has been experiencing a rapid urbanization process, which may lead to an increase in unsustainable land use and urban poverty. Assessing the spatiotemporal characteristics of urbanization dynamics is especially important and needed for the sustainable development of Africa. Satellite-based nighttime light (NTL) data are widely used to monitor the dynamics of urban growth from global to local scales. In this study, urban growth patterns across Africa were analyzed and discussed using stable nighttime light datasets obtained from DMSP/OLS (the Defense Meteorological Satellite Program's Operational Line-scan System) spanning from 1992 to 2013. We partitioned the nighttime lighting areas into three types (low, medium, and high) using thresholds derived from the Brightness Gradient (BG) method. Our results indicated that built-up areas in Africa have increased rapidly, particularly those areas with low nighttime lighting types. Countries with higher urbanization levels in Africa, like South Africa, Algeria, Egypt, Nigeria, and Libya, were leading the brightening trend. The distribution of nighttime lighting types was consistent with the characteristics of urban development, with high nighttime lighting types showed up at the urban center, whereas medium and low nighttime lighting types appeared in the urban-rural transition zone and rural areas respectively. The impacts of these findings on the future of African cities will be further proposed.

**Keywords:** DMSP/OLS; thresholds; nighttime lighting types; urbanization; Africa



## 1. Introduction

Urbanization is often accompanied by growing populations, aggravating socioeconomic activities, and expanding built-up areas and infrastructures [1,2]. Therefore, the rapid rate of urbanization has caused many associated environmental problems, such as air pollution, carbon dioxide emissions, the urban heat island effect, the degradation of natural resources, and the loss of agricultural land [3,4]. To measure and monitor urbanization dynamics from the local to the global scale, it is necessary to obtain a comprehensive understanding of landscape changes and their environmental effects.

Statistical variables, such as socioeconomic and census data, are useful and auxiliary in urbanization studies [5,6]. However, such data are usually absent or unavailable in many less-developed countries to be of help in monitoring urbanization dynamics [7]. In such conditions, satellite data, with various resolutions at multiple spatiotemporal scales, has been extensively applied to study urbanization and its impacts on ecosystems [7–9]. Landsat data were used to analyze urban land-use and land-cover change by various researchers [7,10,11]. Moderate Resolution Imaging Spectroradiometer (MODIS) data with a medium resolution of 500 m were also used to study urbanization dynamics and map urban sprawl [4,12]. These satellite products are mainly applied to monitor land-use and

land-cover change in urban areas. Compared with other satellite products, satellite-based nighttime light (NTL) data can be quantitatively associated with demographic and socioeconomic variables, and can also be applied to monitor the urbanization process [13,14].

NTL data derived from the Operational Line-scan System (OLS) of the Defense Meteorological Satellite Program (DMSP), which excludes glare and sunlight effects, can provide explicit information on artificial lighting during the night [15]. The nocturnal observation of artificial lighting is mainly associated with human activities and contemporary urban settlements, including commercial and entertainment activities, transportation corridors, and other elements of the built environment [16]. Many studies have successfully captured the positive correlations between DMSP/OLS NTL data, demographic [17,18] and socioeconomic variables [19,20], and energy consumption [21,22], which are often used as proxies for urban settlements. As a result, it is an efficient way to analyze and monitor the urbanization process using long-term anthropogenic nighttime light satellite datasets. NTL data was first employed to map urban areas as early as 1973 [15]. DMSP/OLS NTL data have been extensively used to map urban extent and study urbanization dynamics from global to local scales [9,23–28]. Several built-up area extraction methods have been proposed, such as thresholding [6,13], neighborhood statistics analysis (NSA) method [29], and the image classification method [30]. Thresholding is the most commonly used method in NTL-based urban extraction studies. Various thresholding methods have been proposed by many studies, such as empirical thresholding (ET) [31,32], local-optimized thresholding (LOT) [33–35], and brightness gradient-based thresholding (BG) [9]. Of these methods, ET is the simplest but may lead to overestimation or underestimation in urban extraction, while LOT is more accurate but can only be used at a local scale. Compared with other thresholding methods, BG can reduce the effect of overestimation and underestimation, and can be used on a larger scale. BG was first proposed by Ma et al. (2015) [9] in urban areas in China in 2015. The approach classified the NTL images into five different urban categories for 274 Chinese cities based on the quadratic polynomial relationship between the digital number (DN) value and the corresponding gradient brightness of each pixel. The BG method can better reflect urbanization dynamics in spatial and temporal dimensions. Based on the BG method, Kamarajugedda et al. (2017) [36] delineated 15 major Southeast Asian cities into three urban categories and analyzed the urbanization dynamics of these cities. Zhao et al. (2018) [28] applied the BG method to study the spatiotemporal characteristics of the urbanization dynamics of 11 Southeast Asian countries and the relationship between NTL types and human settlements.

Today the world's large cities are generally concentrated in the global South (i.e., Africa). The UN has predicted that between 2018 and 2050, the global urban population will grow by 2.5 billion, with nearly 90% of the increase in Asia and Africa [37]. Most researchers focused on the urbanization dynamics of China, the US, Asia, and Europe [7,9,26,29,38], but limited studies have been found for other regions, such as Latin America and Africa. Currently, most of the existing studies on African urbanization are case studies. For example, Chai et al. (2019) [39] studied small urban settlements in Nigeria and the Democratic Republic of the Congo by combining time series of Landsat and nighttime light satellite data. Using the data from The Atlas of Urban Expansion (2016 Edition), Xu et al. (2019) [40] quantified 25 African cities' growth and form changes by analyzing the spatiotemporal dynamics of urban land densities over three-time points (1990, 2000, and 2014). Ren et al. (2020) [41] studied informal settlements in Nairobi, Kenya using the spatial population data of 2000, 2010, and 2020. A comprehensive and consecutive understanding of different urbanization dynamics across Africa is still lacking.

The main objective of this research is to analyze the spatiotemporal characteristics of urban growth across Africa using DMSP/OLS NTL data from 1992 to 2013. Firstly, we calibrated the DMSP/OLS NTL data through four steps, including intercalibration, intra-annual composition, inter-annual series correction, and enhancing variability through Normalized Difference Vegetation Index (NDVI) data. Secondly, the BG method proposed by Ma et al. (2015) [9] was used to establish a quadratic relationship between the brightness

gradient and the DN value of each pixel. Then, the nighttime lighting areas of Africa were classified into three categories (low, medium, and high) during the period 1992–2013. Finally, the spatiotemporal characteristics of urban growth from 1992 to 2013 across Africa were measured and discussed.

## 2. Study Areas and Data

### 2.1. Study Areas

Africa is the second largest continent with a total area of about 32 million km². Based on geographic location, Africa can be split into Northern Africa, Western Africa, Southern Africa, Eastern Africa, and Middle Africa (Figure 1). A large number of people are continuing to migrate to cities, which implies the extension of these cities. The UN has forecasted that the urban population in Africa will increase by 172% from 2018 to 2050 [37]. As the least urbanized continent, the urbanization rate of Africa is much faster than the rest of the world, which has brought forward huge demands and challenges to urban planning and management. The effective control and governance of African cities is related to sustainable development for Africa and even for the world.

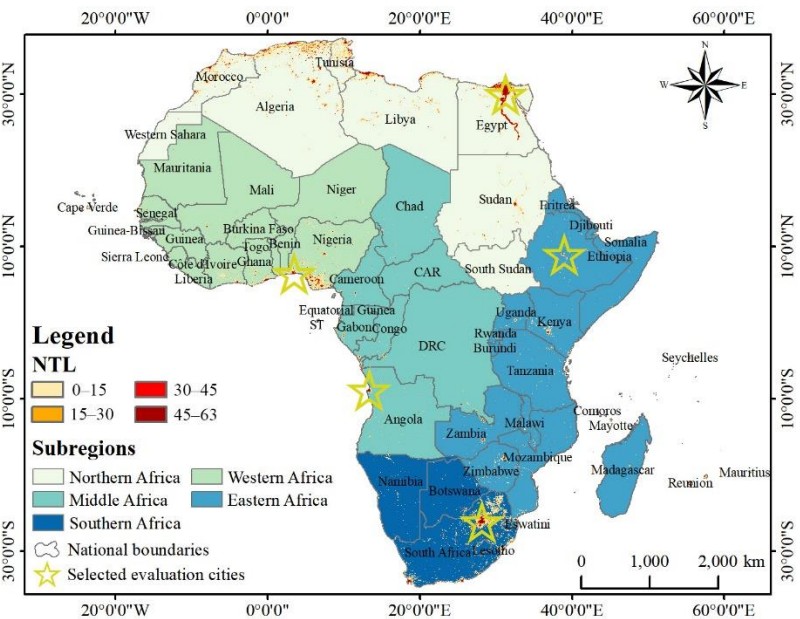

**Figure 1.** Nighttime light (NTL) image in 2013, of subregions and selected evaluation cities for Africa.

### 2.2. Data Source

The National Geophysical Data Center (NGDC) of National Oceanic and Atmospheric Administration (NOAA) released the DMSP/OLS nighttime data for the years 1992 to 2013. The cloud-free stable light imagery, with DN value ranging from 0 to 63, from Version 4 is the most commonly used nighttime light product. The spatial resolution of this nighttime product is 30 arc seconds (approximately 1 km). There are 34 DMSP/OLS nighttime images from six sensors that can be obtained from the website https://www.ngdc.noaa.gov/eog/download.html, including F101992-F101994, F121994-F121999, F141997-F142003, F152000-F152007, F162004-F162009, and F182010-F182013.

The version 1 (v1) Normalized Difference Vegetation Index (NDVI) dataset of the third generation Global Inventory Modelling and Mapping Studies (GIMMIS) derived from NASA Earth Exchange (NEX), (https://nex.nasa.gov/nex/projects/1349/) was used in this research to improve the variability and reduce the saturation effects of the NTL data. It is a global 15-day NDVI product, which is generated from different Advanced Very High-Resolution Radiometer (AVHRR) sensors. Due to the uniquely long AVHRR record, GIMMIS 3g.v1 NDVI has the highest temporal consistency with a long-time series spanning from 1981 to 2015 [42]. The spatial resolution of this NDVI data is 1/12 arc

degrees (approximately 8 km). In this study, the annual mean NDVI was obtained by averaging all the data in the same year, resampled to 1 km.

The reference land-use maps and arterial roads of five selected evaluation cities (Figure 1) were collected from *The Atlas of Urban Expansion (2016 Edition)* (http://www.atlasofurbanexpansion.org) circa 2013. The land-use maps were derived from the Landsat imagery with a spatial resolution of 30 m, which was classified into three classes: built-up areas, open space, and water bodies, through unsupervised classification methods [43].

The administrative boundary used in this study was from the website of Database of Global Administrative Areas (GADM) (http://gadm.org/country). To remove the highlighted pixels of water, a water mask from the research results of Pekel et al. (2016) [44] was adopted. Moreover, the population density in 2013 was obtained from the LandScan (https://landscan.ornl.gov/).

## 3. Methods

### 3.1. Calibration

DMSP/OLS NTL data lack continuity and comparability due to no onboard calibrations, different sensors, and differences in passing time. To eliminate the different responses among sensors and inter-annual variations, the calibration steps of DMSP/OLS NTL data have been carried out as follows.

### 3.1.1. Inter-Calibration

The modified invariant region (MIR) method was first introduced by Elvidge et al. (2009) [14] and has been widely used by other researchers [28,33,45]. Referencing to the MIR method, Sicily as the invariant target areas and the F182010, which has the highest digital values, as the reference image was selected to inter-calibrate all the 34 satellite images from DMSP/OLS in Africa. The quadratic polynomial model was selected in this study:

$$DN_1 = a \times DN_0^2 + b \times DN_0 + c \qquad (1)$$

$DN_0$ is the original $DN$ value, $DN_1$ is the inter-calibrated $DN$ value, and a, b, c are all coefficients.

### 3.1.2. Intra-Annual Composition

There are two annual composites for 1994 and 1997–2007. To fully use the information of the two sensors for the same year, an intra-annual composition by averaging DN values of the pixels of two NTL images was performed as follows:

$$DN_2 = \begin{cases} 0, & DN_1^a = 0 \big| DN_1^b = 0 \\ \left(DN_1^a + DN_1^b\right)/2, & otherwise \end{cases} \qquad (2)$$

$DN_1^a$ and $DN_1^b$ are $DN_1$ values of the two NTL images for the same year, respectively. $DN_2$ is the DN value after intra-annual composition.

### 3.1.3. Inter-Annual Series Correction

The objective of this step was to remove inconsistencies and to correct DN values for consistent pixels. Considering the characteristics of the urbanization process, it is assumed that lighted pixels that appeared in the earlier NTL image should be preserved in the later NTL image, and the DN values of the earlier NTL pixels should not be greater than the later NTL pixels [46]. Therefore, we corrected the NTL data as follows:

$$DN_3 = \begin{cases} DN_{2(n-1,i)}, & DN_{2(n-1,i)} > DN_{2(n,i)} \\ DN_{2(n,i)}, & otherwise \end{cases} \qquad (3)$$

$DN_{2(n-1,i)}$ and $DN_{2(n,i)}$ are the DN values of the NTL data after intra-annual composition in the n-1th and the nth year, respectively. $DN_3$ denotes the DN value after inter-annual series correction.

### 3.1.4. Enhancement of Variability

As the maximum DN value goes up to 63, there is a saturation effect in the DMSP/OLS NTL data. To reduce such effect, NDVI was applied to correct the high-value pixels and enhance the variability based on the hypothesis that there is a highly negative relationship between the vegetation abundance (NDVI) and impervious surfaces (DN values) in urban areas. Referencing to the Vegetation Adjusted NTL Urban Index (VANUI) proposed by Zhang et al. (2013) [47], the last calibration step is as follows:

$$DN_4 = (1 - NDVI) \times DN_3 \tag{4}$$

where $DN_4$ is the DN value of the NTL data after NDVI calibration. NDVI is the annual mean GIMMIS 3g.v1 NDVI data. The negative NDVI values are abandoned, as they are usually connected with clouds, water, and glacier. As a result, the NDVI values are constrained to the range between 0 and 1.0.

### 3.2. The Spatial Gradient of Night-Time Lights

According to the BG method proposed by Ma et al. (2015) [9], the pixel-level brightness gradient (BG) of the central grid cell was calculated as Equation (5), which was used to reveal the spatial changes for the NTL data. Equations (6) and (7) were explanatory and supplementary equations for Equation (5). The illustration is shown in Figure 2, in which $DN_4$ was the central grid cell and the BG value was measured by its eight neighboring grid cells, including $DN_0$, $DN_1$, $DN_2$, $DN_3$, $DN_5$, $DN_6$, $DN_7$, and $DN_8$.

$$BG = \sqrt{(dDN/dx)^2 + (dDN/dy)^2} \tag{5}$$

$$dDN/dx = [(DN_2 + 2DN_5 + DN_8) - (DN_0 + 2DN_3 + DN_6)]/8 \tag{6}$$

$$dDN/dy = [(DN_6 + 2DN_7 + DN_8) - (DN_0 + 2DN_1 + DN_2)]/8 \tag{7}$$

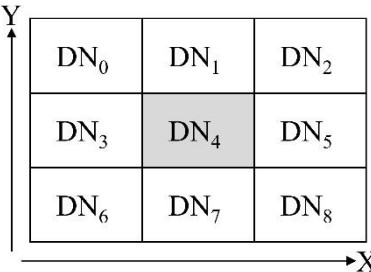

**Figure 2.** Schematic diagram of the calculation of the brightness gradient.

### 3.3. Extracting Different Urban Categories

According to Ma et al. (2015) [9], the low BG values of pixels were found in urban centers with high DN values and rural areas with low DN values (Figure 3a,c), while the high BG values showed up in the transition zone between urban and rural with medium DN values (Figure 3b,d). The relationship between the DN value and the BG of a pixel can be modeled as Equation (8) (Figure 3e,f).

$$BG = a \times DN^2 + b \times DN + c \tag{8}$$

*BG* denotes the brightness gradient, *DN* is the calibrated *DN* value, and a, b, c are all coefficients.

This BG method has been applied to identify the urbanization dynamics in China [9] and South Asia [28,36]. Based on the current situation of African cities, the pixels of the African NTL image were partitioned into three different categories: low (from $DN_0$ to $DN_1$), medium (from $DN_1$ to $DN_2$), and high (from $DN_2$ to $DN_3$). The detailed information can be seen in Table 1 and Figure 3g. Figure 3h took Addis Ababa city as an example to show the partitioning result.

**Table 1.** Calculations for the split points in Figure 3.

| Point | Digital Number (DN) | Brightness Gradient (BG) |
|---|---|---|
| $P_0(DN_0, BG_0)$ | $DN_{min}$ | $aDN_{min}^2 + bDN_{min} + c$ |
| $P_1(DN_1, BG_1)$ | $-\frac{b}{2a} - \sqrt{\frac{BG_1-c}{a} + \frac{b^2}{4a^2}}$ | $\frac{BG_0+BG_2}{2}$ |
| $P_2(DN_2, BG_2)$ | $-\frac{b}{2a}$ | $\frac{4ac-b^2}{4a}$ |
| $P_3(DN_3, BG_3)$ | $DN_{max}$ | $aDN_{max}^2 + bDN_{max} + c$ |

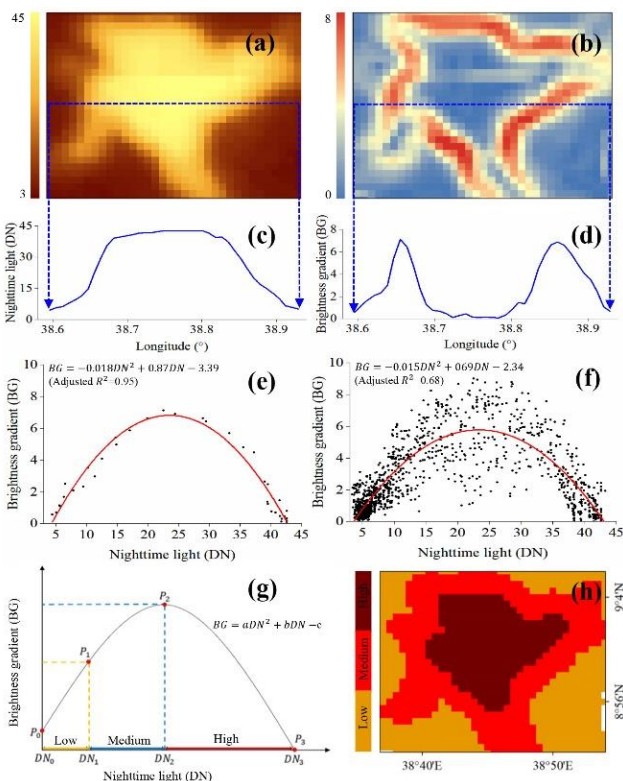

**Figure 3.** (**a**) Defense Meteorological Satellite Program's Operational Line-scan System (DMSP/OLS) nighttime light and (**b**) brightness gradient in 2013 for Addis Ababa city; (**c**) and (**d**) show the latitudinal transect of the pixel-level nighttime light and brightness gradient across Addis Ababa city, respectively; (**e**) and (**f**) the quadratic regression relationship between nighttime light and brightness gradient in 2013 for the latitudinal transect and Addis Ababa city; (**g**) schematic diagram of the relationship between nighttime light and brightness gradient and the subdivision based on the Equation (8); (**h**) results of partitioning for Addis Ababa city.

## 4. Results

### 4.1. Evaluation of NTL Data Calibration Results

It took four steps to calibrate the raw NTL data for Africa from 1992 to 2013, including inter-calibration, intra-annual composition, inter-annual correction, and enhancement of variability using NDVI data. Figure 4a shows the total DN values of the original NTL data before calibration, from which we can see that the raw NTL data lacks continuity and comparability. The coefficients and $R^2$ of the inter-calibration model can be seen in Table 2, and the results are presented in Figure 4b. The values of the $R^2$ are all higher than

0.8 confirming the reliability of the inter-calibration model. Figure 4c–e show the total DN values after the intra-annual composition, inter-annual series correction, and enhancement of variability, respectively. As shown in Figure 4c, the discrepancies in the same year's DN values between the two satellites were reduced. As Figure 4d,e demonstrate, the abnormal volatility of DN values between two adjacent years was minimized. The continuity, comparability, and variability of the NTL data were significantly improved after the calibration.

**Table 2.** Adjusted parameters based on Equation (1) in Sicily using F182010.

|  | Year | a | b | c | $R^2$ |
|---|---|---|---|---|---|
| F10 | 1992 | −0.0131 | 1.7757 | 0.6679 | 0.8754 |
|  | 1993 | −0.0172 | 2.0087 | −0.8980 | 0.8558 |
|  | 1994 | −0.0143 | 1.8252 | 0.9013 | 0.9081 |
|  | 1994 | −0.0070 | 1.4134 | 1.3803 | 0.8997 |
|  | 1995 | −0.0113 | 1.6671 | −0.2685 | 0.9176 |
| F12 | 1996 | −0.0104 | 1.6066 | 1.1547 | 0.9057 |
|  | 1997 | −0.0099 | 1.5971 | −0.3859 | 0.9128 |
|  | 1998 | −0.0085 | 1.4840 | −0.3349 | 0.9119 |
|  | 1999 | −0.0060 | 1.3057 | 1.1007 | 0.8693 |
|  | 1997 | −0.0191 | 2.1265 | −0.2056 | 0.8968 |
|  | 1998 | −0.0163 | 1.9091 | 1.6969 | 0.8604 |
|  | 1999 | −0.0165 | 1.9589 | −0.1298 | 0.8915 |
| F14 | 2000 | −0.0139 | 1.7967 | 0.9982 | 0.9192 |
|  | 2001 | −0.0138 | 1.8103 | −0.1493 | 0.9223 |
|  | 2002 | −0.0108 | 1.6010 | 1.1430 | 0.9109 |
|  | 2003 | −0.0135 | 1.7803 | −0.0531 | 0.9100 |
|  | 2000 | −0.0095 | 1.5496 | −1.2544 | 0.8761 |
|  | 2001 | −0.0089 | 1.5291 | −0.9127 | 0.8850 |
|  | 2002 | −0.0065 | 1.3676 | 0.0188 | 0.8904 |
| F15 | 2003 | −0.0170 | 1.9969 | −0.0626 | 0.9000 |
|  | 2004 | −0.0127 | 1.7318 | 1.2487 | 0.9153 |
|  | 2005 | −0.0127 | 1.7456 | 0.2670 | 0.8886 |
|  | 2006 | −0.0133 | 1.7953 | 0.1818 | 0.9352 |
|  | 2007 | −0.0134 | 1.7960 | 1.0956 | 0.9520 |
|  | 2004 | −0.0104 | 1.5961 | 0.4025 | 0.9025 |
|  | 2005 | −0.0158 | 1.9382 | −0.8261 | 0.9100 |
| F16 | 2006 | −0.0093 | 1.5785 | −0.1035 | 0.9330 |
|  | 2007 | −0.0062 | 1.3538 | 0.0904 | 0.9197 |
|  | 2008 | −0.0073 | 1.4157 | 0.2696 | 0.9078 |
|  | 2009 | −0.0074 | 1.4194 | 1.6195 | 0.9487 |
|  | 2010 | 0 | 1 | 0 | 1 |
| F18 | 2011 | −0.0037 | 1.1454 | 1.3936 | 0.8936 |
|  | 2012 | −0.0021 | 1.0878 | 0.4981 | 0.9295 |
|  | 2013 | −0.0035 | 1.1469 | 0.7597 | 0.9247 |

Figure 5 reveals a pixel-level DN value comparison between nighttime light before NDVI-induced calibration (Figure 5a,c) and after the calibration (Figure 5b,d) across a latitudinal transect (Figure 5e,f) of Johannesburg and Cairo. It is clear that the effects of NTL saturation were reduced and the variation of the NTL signals within urban areas were increased after the enhancement of Variability by NDVI.

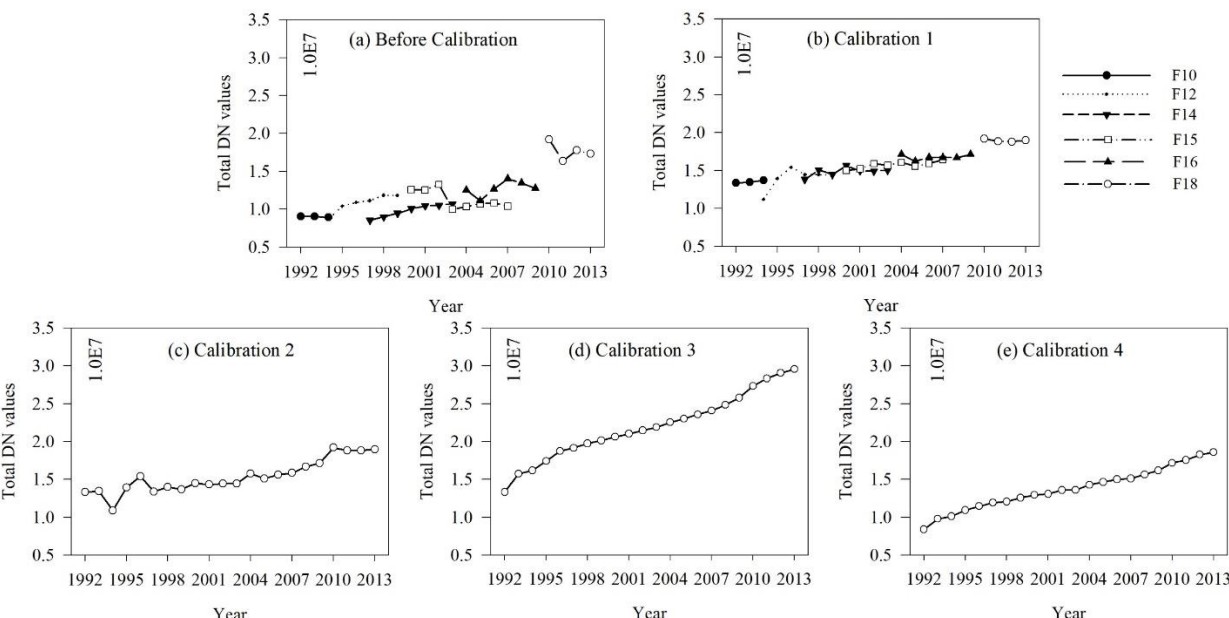

**Figure 4.** Comparisons of the total DN values at different calibration stages for Africa: (**a**) Original NTL data; (**b**) Inter-calibration; (**c**) Intra-annual composition; (**d**) Inter-annual series correction; (**e**) Enhancement of variability.

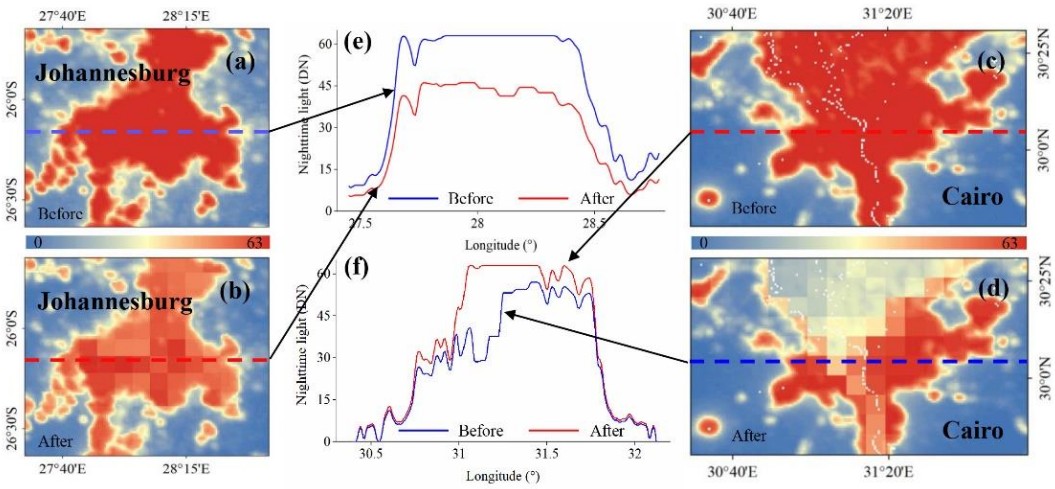

**Figure 5.** (**a**,**c**) Nighttime light before NDVI-induced calibration of Johannesburg and Cairo, respectively; (**b**,**d**) after the enhancement of variability using NDVI over Johannesburg and Cairo, respectively; (**e**,**f**) latitudinal transects of nighttime light across Johannesburg and Cairo, respectively.

*4.2. Long-Term Spatiotemporal Trends of Different Nighttime Lighting Types at Different Scales*

4.2.1. Continental and Sub-Regional Level Trends in Different Nighttime Lighting Areas

Depending on the relationship between the DN value and the corresponding brightness gradient of pixels, two thresholds were obtained and used to classify the lighted areas of 22 nighttime images into three types. Figure 6a demonstrates the long-term trends of different nighttime lighting types across Africa from 1992 to 2013. It can be seen that all of the nighttime lighting types have a growing trend. The growth of the low nighttime lighting types was the fastest at a rate of 41917 km$^2$·year$^{-1}$, followed by the medium nighttime lighting types with an increase of 7108 km$^2$·year$^{-1}$. The high nighttime lighting types grew much more slowly than the other two types, with a growing rate of 2059 km$^2$·year$^{-1}$. The nighttime lighting area (NLA) in 2013 was 2.7 times that of 1992, which increased from $0.7 \times 10^6$ km$^2$ in 1992 to $1.9 \times 10^6$ km$^2$ in 2013. The annual growth rate of NLA was 4.9%,

which was relatively close to the annual growth rate of built-up areas (nearly 5%) in Africa during 1990–2014 [40]. The increase of African NLA is $1.2 \times 10^6$ km$^2$, which is equivalent to 4.8 times South Africa's NLA in 2013.

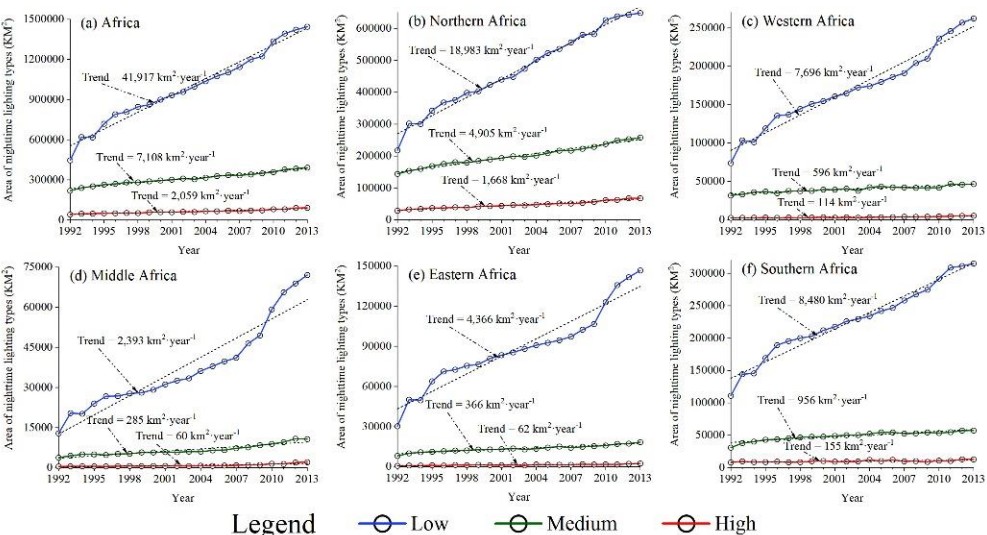

**Figure 6.** Long-term trends of different nighttime lighting types from 1992 to 2013: (**a**) Africa; (**b**) Northern Africa; (**c**) Western Africa; (**d**) Middle Africa; (**e**) Eastern Africa; (**f**) Southern Africa.

Urbanization levels show a great diversity among the five sub-regions due to the different geographic environment, socioeconomics, history, and state conditions. The spatiotemporal trends of three different nighttime lighting types in five sub-regions (Northern, Western, Middle, Eastern, and Southern Africa) are presented in Figure 6b–f, respectively. The low nighttime lighting types increased the most quickly in five sub-regions. At the same time, the other two types grew much more slowly compared to the low nighttime lighting types. The low nighttime lighting types changed most quickly in Northern Africa with an increasing area of 18982.8 km$^2 \cdot$year$^{-1}$, followed by Southern Africa (8479.6 km$^2 \cdot$year$^{-1}$), Western Africa (7695.6 km$^2 \cdot$year$^{-1}$), Eastern Africa (4366.4 km$^2 \cdot$year$^{-1}$), and Middle Africa (2393.4 km$^2 \cdot$year$^{-1}$). As for the medium nighttime lighting areas, growth in Northern Africa (4904.8 km$^2 \cdot$year$^{-1}$) accounts for more than half of Africa's total growth (7107.6 km$^2 \cdot$year$^{-1}$). At the same time, the growth of high nighttime lighting types also happened mostly in Northern Africa, which made up 81% of the total increase of high nighttime lighting types across Africa. As for the annual growth rate of the NLA in five sub-regions, Middle Africa had the highest annual growth rate of 7.9%, followed by Eastern Africa (7.2%), Western Africa (5.2%), Southern Africa (4.6%), and Northern Africa (4.4%), while the annual growth rate for the whole of Africa was 4.9%.

### 4.2.2. Leading Countries for Different Nighttime Lighting Types

It can be seen from the data in Figure 7 and Table 3 that South Africa, Algeria, Nigeria, Egypt, and Libya have led the African brightening trend both in 1992 and 2013. South Africa had the largest nighttime lighting areas (hereafter, NLA) both in 1992 ($10.3 \times 10^4$ km$^2$) and 2013 ($25.3 \times 10^4$ km$^2$), accounting for 19.7% and 16.9% of the total NLA in Africa, respectively. In 1992, the NLA in the top five countries accounted for 76.8% of the total NLA in Africa, and the proportion fell to 58.8% in 2013. As for the increased NLA between 1992 and 2013, Morocco ranked third, surpassing Egypt and Nigeria. The increased NLA in the top six countries made up 56.2% of the total increased NLA in Africa. Of the five leading countries, three were in Northern Africa, one in Southern Africa, and one in Western Africa.

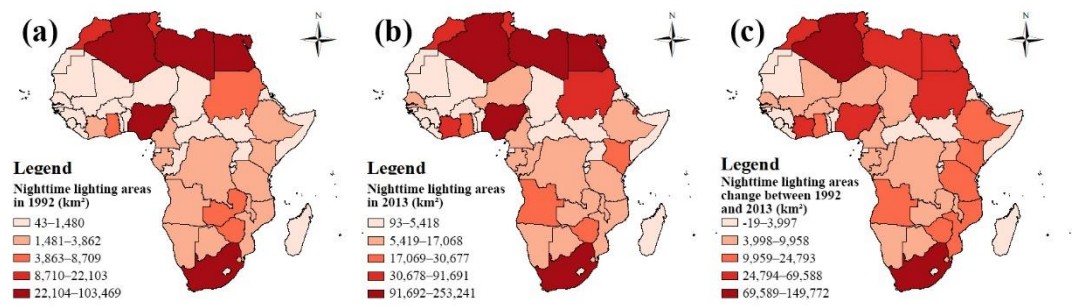

**Figure 7.** Nighttime lighting areas: (**a**) in 1992; (**b**) in 2013; (**c**) change between 1992 and 2013.

**Table 3.** Statistics of NLA in major countries from 1992 to 2013.

| Rank | 1992 | | 2013 | | 1992–2013 | |
|------|------|------|------|------|------|------|
| | NLA ($\times 10^4$ km$^2$) | Ratio (%) * | NLA ($\times 10^4$ km$^2$) | Ratio (%) ** | Change Areas ($\times 10^4$ km$^2$) | Ratio (%) *** |
| 1 | South Africa (10.3) | 19.7% | South Africa (25.3) | 16.9% | South Africa (15.0) | 15.4% |
| 2 | Algeria (10.2) | 19.4% | Algeria (24.1) | 16.1% | Algeria (13.9) | 14.3% |
| 3 | Nigeria (6.9) | 13.2% | Nigeria (13.1) | 8.8% | Morocco (7.0) | 7.2% |
| 4 | Libya (6.6) | 12.5% | Egypt (12.9) | 8.6% | Egypt (6.6) | 6.8% |
| 5 | Egypt (6.3) | 12.0% | Libya (12.5) | 8.4% | Nigeria (6.2) | 6.4% |
| 6 | Morocco (2.2) | 4.2% | Morocco (9.2) | 6.1% | Libya (5.9) | 6.1% |
| 7 | Tunisia (1.9) | 3.6% | Tunisia (6.6) | 4.4% | Tunisia (4.7) | 4.9% |
| 8 | Zimbabwe (0.9) | 1.6% | Côte d'Ivoire (4.7) | 3.2% | Côte d'Ivoire (4.4) | 4.5% |
| 9 | Sudan (0.8) | 1.4% | Djibouti (4.7) | 3.1% | Djibouti (4.4) | 4.5% |
| 10 | Ghana (0.6) | 1.1% | Sudan (4.4) | 3.0% | Sudan (3.6) | 3.8% |
| Africa | 52.5 | 100% | 149.6 | 100% | 97.1 | 100% |

Note: NLA represents nighttime lighting areas; * the ratio of NLA in each country in Africa in 1992; ** the ratio of NLA in each country in Africa in 2013; *** the ratio of increased NLA in each country in Africa between 1992 and 2013.

We also calculated the average annual growth rate (hereafter, AGR) of total nighttime light and three different nighttime lighting types at the country scale from 1992 to 2013, as shown in Figure 8. The average annual growth rate of different nighttime lighting types in African countries showed significant differences during the period 1992 to 2013. The spatial distribution of the AGR for total nighttime light (Figure 8a) and low nighttime lighting type (Figure 8d) were similar, as AGR in the top five countries (South Africa, Algeria, Nigeria, Egypt, and Libya), Zimbabwe, Zambia, and Togo was low, while in other countries it was relatively higher. This was mainly because the low nighttime lighting type was the most common in NTL data, dominating the change in total nighttime light. The spatial distribution of the AGR for high nighttime lighting type (Figure 8b) was opposite to that for the total nighttime light and low nighttime lighting type, to some extent. The AGR for high nighttime lighting type in some countries was very low, especially in most countries in Middle Africa. As for the AGR spatial distribution for medium nighttime lighting type (Figure 8c), the difference was relatively small compared with other nighttime lighting types.

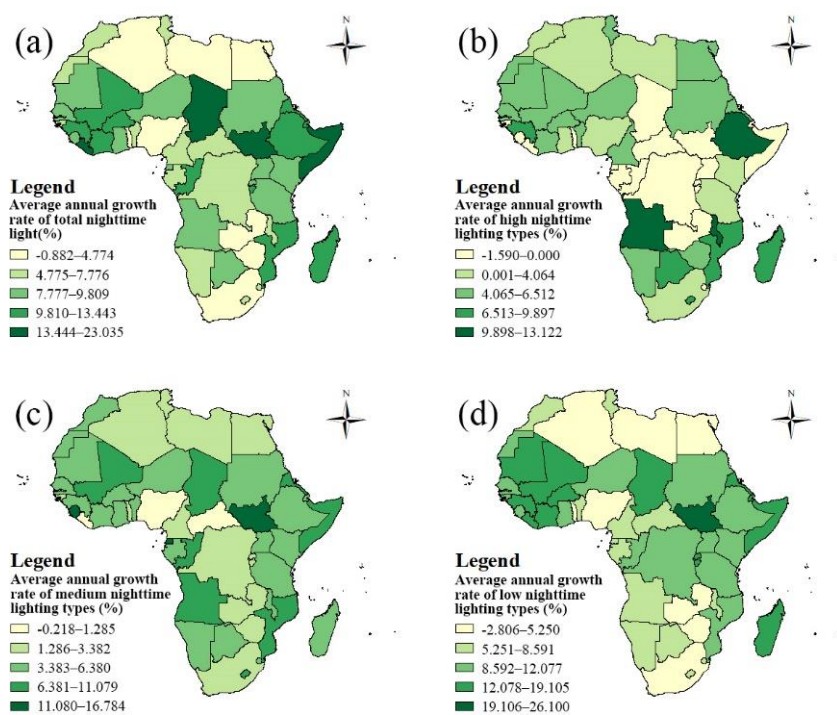

**Figure 8.** Average annual growth rate of: (**a**) total nighttime light; (**b**) high nighttime lighting types; (**c**) medium nighttime lighting types; (**d**) low nighttime lighting types.

### 4.3. Temporal and Spatial Transitions of Different Nighttime Lighting Types

Long-term changes in different nighttime lighting types can be used to explain the trends and variations of the urbanization process along with diversified human activities. The transitions of different nighttime lighting types across Africa between 1992 and 2013 are summarized in Table 4. It can be seen from the data in Table 4 that the total pixels of Africa were 29,902,494. There were 1,451,048 lighted pixels in 2013, which accounted for 4.9% of the total area in Africa. Among the lighted pixels in 2013, 75.7% were low nighttime lighting areas, 19.7% were medium nighttime lighting areas and only 4.5% were high nighttime lighting areas. For the high nighttime lighting areas in 2013, 41.6% and 45.4% of them came from high and medium nighttime lighting areas in 1992, respectively, whereas only 7.8% were from areas lighted after 1992. As for the medium nighttime lighting areas in 2013, 44.4% were sourced from unchanged medium nighttime lighting areas from 1992, while 33.8% and 21.8% were from low nighttime lighting areas and areas lighted after 1992, respectively. For the low nighttime lighting areas, 78.9% were from areas lighted after 1992, and 21.1% came from unchanged low nighttime lighting areas from 1992. There were 934,405 pixels lit after 1992, which accounts for 64.6% of the total lighted pixels in 2013. In summary, these results showed that many unlighted areas in Africa were rapidly transformed into lighted areas, especially with low nighttime lighting types, during the period 1992~2013. The source of each nighttime lighting type mainly came from the same nighttime lighting types and the darker types.

**Table 4.** Transition matrix of different nighttime lighting types between 1992 and 2013 across Africa.

| 1992 | | N | L | M | H | Total |
|---|---|---|---|---|---|---|
| | N | 28,451,446 | - | - | - | 28,451,446 |
| | L | 866,744 | 231,942 | - | - | 1,098,686 |
| | M | 62,517 | 96,845 | 127,134 | - | 286,496 |
| **2013** | H | 5144 | 3433 | 29,893 | 27,396 | 65,866 |
| | Total | 29,385,851 | 332,220 | 157,027 | 27,396 | 29,902,494 |

The process of urbanization is generally simultaneous with nighttime light change. Figure 9 provides the spatial transitions of different nighttime lighting types in the selected evaluation cities between 1992 and 2013 to depict the spatial transition pattern at the local scale. Detailed information and the location of five selected cities can be seen in Table 5 and Figure 1, respectively. It is evident that the core urban areas with dense human activities show high nighttime lighting types, whereas the peripheral areas exhibit low nighttime lighting types. The transitions between medium and high nighttime light were close to the urban center. At the same time, the transitions between low and medium were near the urban fringe. Along with the urbanization process, urban areas could potentially become lighter in the next stages. These results indicate that the long-term transition of nighttime lighting types could display the spatial and temporal urbanization process. As shown in Figure 9, the urban expansion of port cities, like Lagos and Luanda, was limited by the terrain and coastline when considering city centers along the coast. In the meantime, the urban core areas of inland cities, for instance, Cairo, Addis Ababa, and Johannesburg, were in the center. The transitions of nighttime lighting types were consistent with urban development characteristics.

**Table 5.** Detailed information of the cities presented in Figure 9.

| Code | Study Areas | Country | Sub-Region | Scale | Types | Population Circa 2013 |
|------|-------------|---------|------------|-------|-------|-----------------------|
| a | Cairo | Egypt | Northern Africa | capital; largest city | inland city | 15,734,934 |
| b | Lagos | Nigeria | Western Africa | largest city | port city | 11,008,356 |
| c | Luanda | Angola | Middle Africa | capital; largest city | port city | 5,555,024 |
| d | Addis Ababa | Ethiopia | Eastern Africa | capital; largest city | inland city | 3,009,130 |
| e | Johannesburg | South Africa | Southern Africa | largest city | inland city | 8,000,158 |

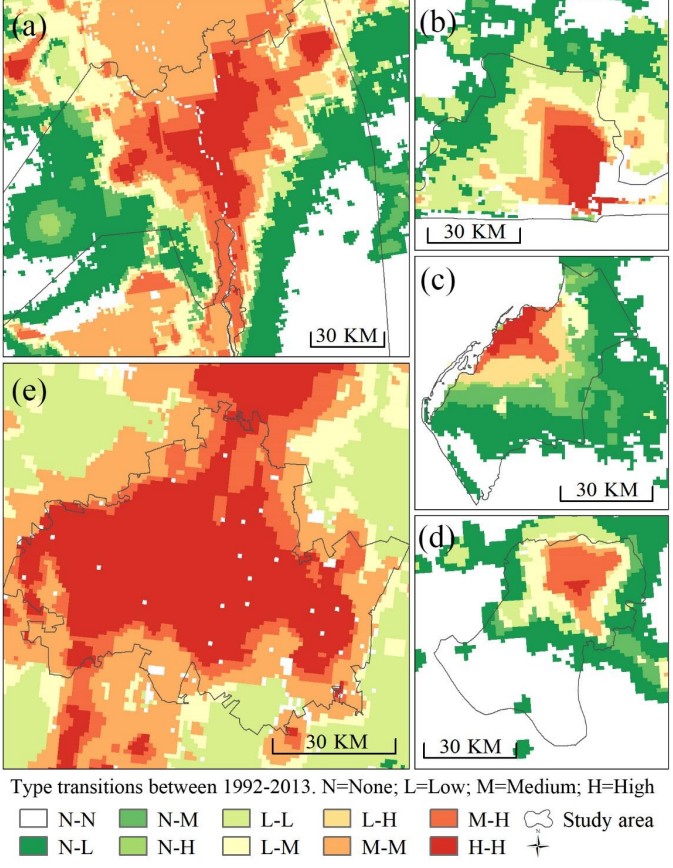

**Figure 9.** Spatial transitions of different nighttime light types in several African cities between 1992 and 2013: (**a**) Cairo; (**b**) Lagos; (**c**) Luanda; (**d**) Addis Ababa; (**e**) Johannesburg.

## 5. Discussion

### 5.1. Relationship between Nighttime Light and Urbanisation

The regional NLA is closely related to urban areas, infrastructure, and population density. The land-use maps from *The Atlas of Urban Expansion (2016 Edition)* (Figure 10a) were used as reference maps of urban extent to compare with the nighttime lighting types derived from the BG results (Figure 10b). As shown in Figure 10a,b, the high nighttime lighting types appeared most often in the central region of urban areas, while the medium nighttime lighting types showed up in the transition zones between urban and rural areas, and rural areas were covered by low nighttime lighting types or no light. This is consistent with the results of Small et al. (2011) [48], that the brightest pixels are generally observed in fully developed urban areas, and less brightly lit pixels show up in less densely built areas, while pixels with the lowest brightness levels are related to a variety of sources.

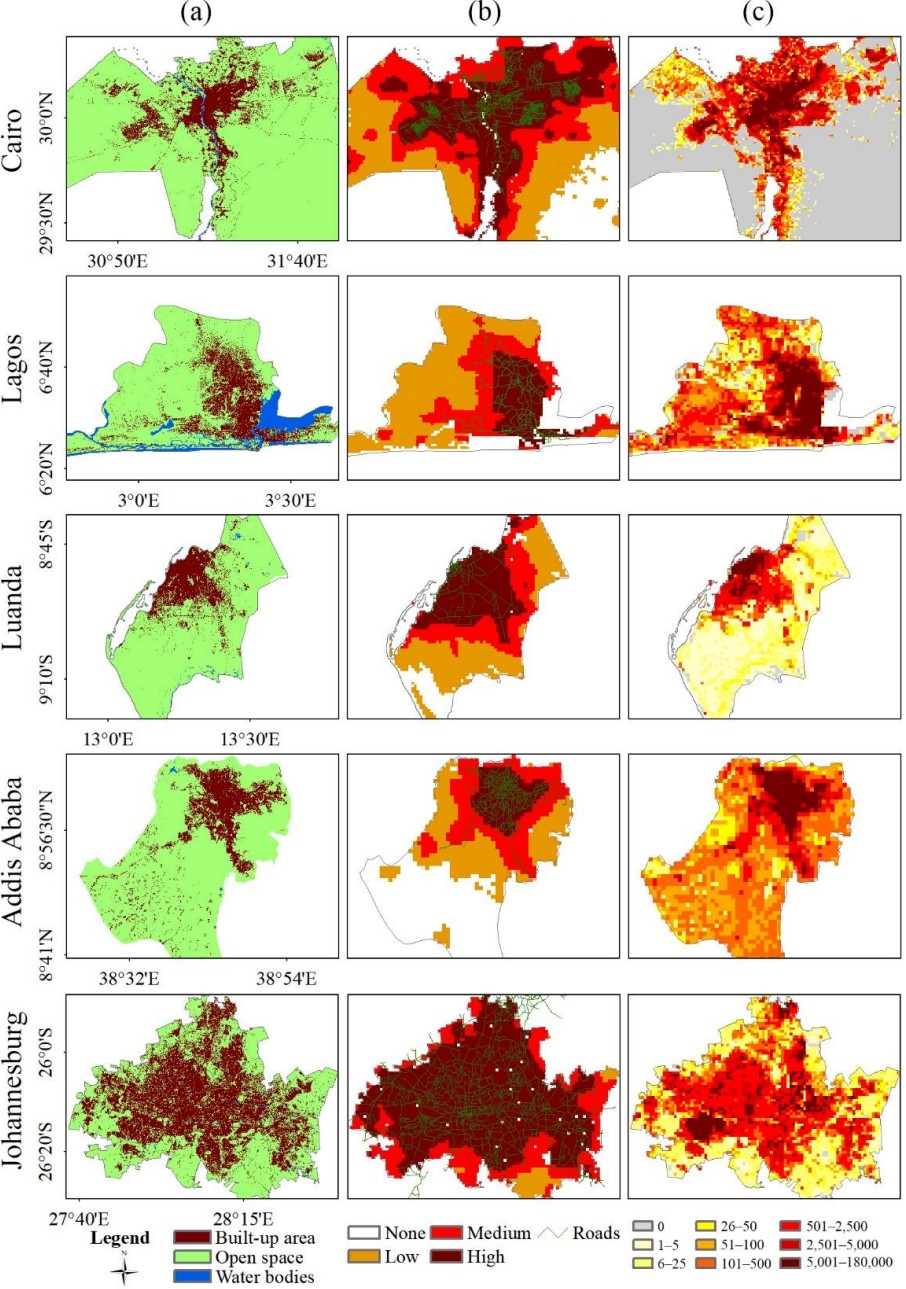

**Figure 10.** Comparative analysis of (**a**) urban land use, (**b**) nighttime lighting areas and arterial roads, and (**c**) population density.

Studies have shown that roads are one of the major sources of nighttime light [49,50]. A comparative analysis between nighttime lighting types and arterial roads was also conducted (Figure 10b). The arterial roads were mainly located in the areas with high nighttime lighting types. Figure 10c exhibits the population distribution for five cities in 2013. Positive spatial correspondence can be visually inspected between high population density and high nighttime lighting types, which verified that DMSP/OLS NTL was related to intensified human activity.

*5.2. Status of Urbanization in Africa*

The main core of the work is based on Africa, which is the least urbanized region in the world, with only 43% of its population living in urban areas. At the same time, Africa has a relatively high urban population growth rate, and there will be 1.5 billion urban dwellers in Africa by 2050 [37], which means there will be a rapid urbanization process from 2018 to 2050 in Africa. Previous studies have highlighted the importance of studying the urbanization process of Africa [39,40,51,52]. Africa is facing the fastest urbanization process in the world and a comprehensive understanding of its urbanization dynamics and patterns is urgently needed within this context.

We have studied urban growth across Africa by partitioning the DMSP/OLS NTL image into three different types from 1992 to 2013. Findings showed that the NTL increased on average by 4.9% per annum, which was close to the study results of Xu et al. (2019) which showed that the average annual growth rate of built-up area was around 5% in 25 African cities during 1990–2014 [40]. In 2010, Southern Africa had the highest urbanization level, followed by Northern Africa, Western Africa, and Middle Africa. Meanwhile, Middle Africa experienced the highest urbanization rate from 2000 to 2010, followed by Western Africa, Eastern Africa, Northern Africa, and Southern Africa [53]. This was similar to our results in Section 4.2. The regions with higher types and larger areas of NTL had a lower average annual growth rate of NTL (or vice versa). Our study indicates that NTL data can be easily used to monitor the urbanization rate in Africa.

Our findings also have implications for sustainable urbanization in Africa. At present, Africa's urbanization faces two fundamental problems: unsustainable land use [54,55] and growing urban poverty [56]. Some studies have pointed out that the expansion of African cities has generally transformed rural areas including farmland into urban areas [57,58]. In 2000, African built-up areas covered 33,025 km$^2$, which will increase by 590% in the following three decades. This urban land cover change rate is the highest in the world [55]. In this study, our results match those observed in earlier studies. The annual growth rate of NTL across Africa was 4.9% during the period 1992–2013. Low nighttime lighting areas increased by 41917 km$^2$ per year, more than four times that of the other two nighttime lighting types (9167 km$^2 \cdot$year$^{-1}$, Figure 6). A large number of unlighted areas are turning into low nighttime lighting areas, which indicates the rapid but unsustainable urbanization in Africa. This process may further induce landscape fragmentation, biodiversity reduction, and greenhouse gas emission [54,55,59].

Another problem with African urbanization is urban poverty. The proportion of the African urban population living on $1.25 a day is nearly 43% [56,60]. Over 70% of the African urban population are living in slums, and new city dwellers are mainly relying on unplanned houses and informal settlements [1,52,53]. Kinshasa, the capital and largest river port of the Democratic Republic of Congo, is also the largest city in Central Africa. As shown in Figure 11, the urban center of Kinshasa was not completely covered by high nighttime light types in 2013, as in Kampala, the capital of Uganda. There were some cities with no high nighttime light types in 2013, for example, Kigali, Arusha, and Nakuru, the urban centers of which were covered by medium nighttime lighting types. The urban fringe of these cities was covered by low nighttime lighting types, and the rural areas were even unlighted. Prior studies have noted that nighttime light can be considered as a proxy for wealth [14,61,62]. It is assumed that the poorly lighted areas with higher population

density have higher percentages of poor people [14]. As mentioned above, urban poverty is common in some African cities.

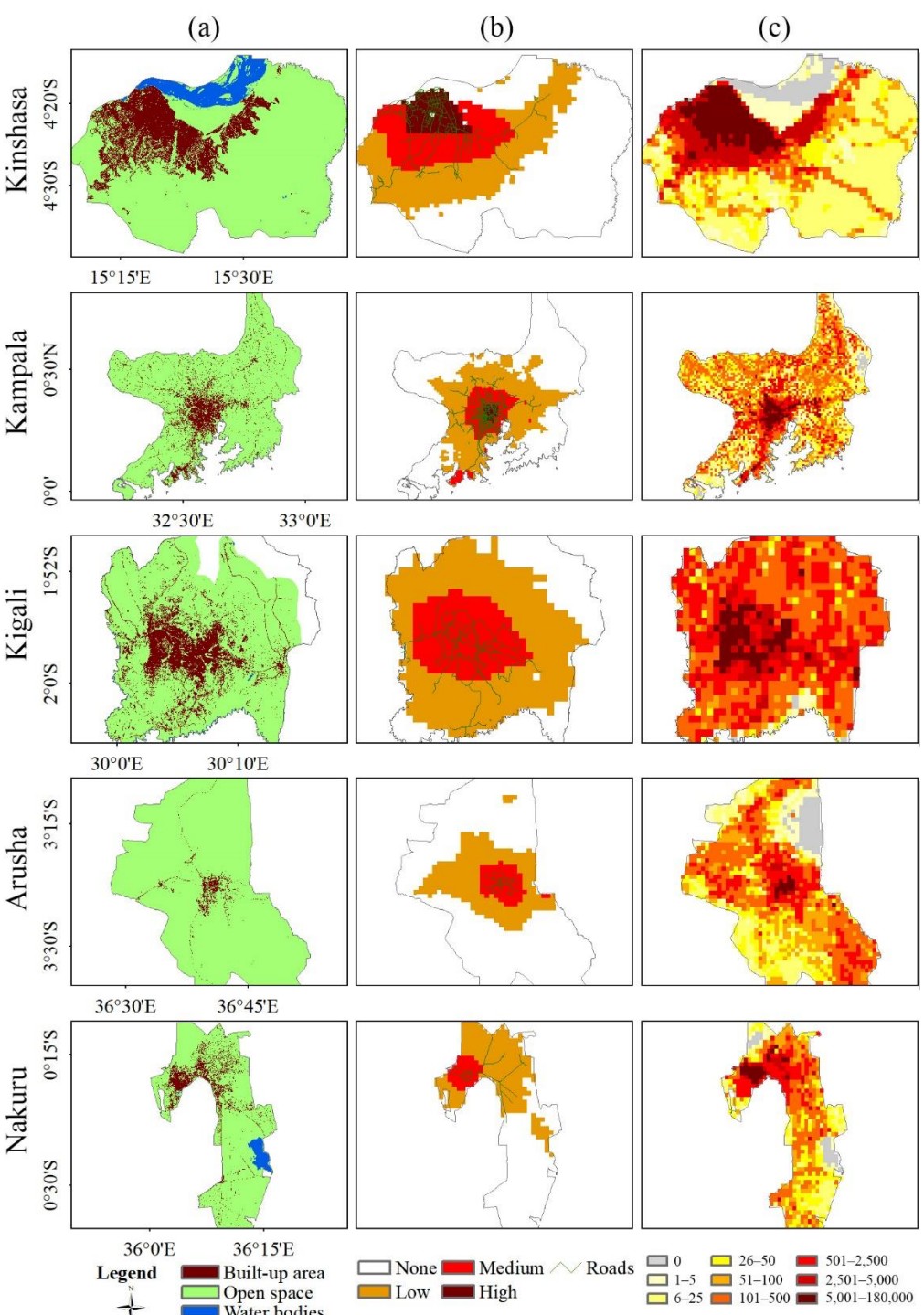

**Figure 11.** Representative poorly lighted cities (**a**) urban land use, (**b**) nighttime lighting areas and arterial roads, and (**c**) population density.

Research suggests that there is a positive relationship between sustainable development and urbanization [63,64]. In the meantime, many studies show that rapid urbanization may have negative effects on urban livelihood with unemployment, poor water supply, and lack of sanitation facilities [60], which would worsen urban poverty conditions [51,54,61,62]. Africa is experiencing a rapid urbanization process nowadays, which poses a major challenge to the sustainable development of Africa. As mentioned above,

Africa is getting brighter, especially those unlighted areas which are turning to low night-time light. These results are in agreement with evidence that the current urbanization rate is already having a bad effect on Africa in terms of urban poverty and unsustainable use of land and energy [53]. Despite this somewhat discouraging argument and reality, some studies suggest that urbanization can promote the sustainable development of Africa, depending on efficient infrastructures and institutions [63–65]. The World Bank regards functional urbanization as a potential driver for African development [66]. A study conducted by Xu et al. (2019) [40] showed that African cities were less compact than cities in other regions and suggested that compact growth policies should be adopted for African sustainable urbanization.

Results from this study provide an insight into the dynamics of urban growth across Africa and the problems of the rapid urbanization. As the second continent, urban sustainability in Africa is very important for the sustainable development of Africa and even for the world. A comprehensive understanding of the complex urbanization process in Africa helps the formulation of better policies. As Africa's urbanization level is estimated to rise, more studies are needed to fully understand cities and urbanization in Africa.

### 5.3. Limitations and Further Research

As NOAA only released the DMSP/OLS NTL data from 1992 to 2013, we only studied the spatiotemporal characteristics of African urbanization in this period. We have noticed that NTL data from VIIRS-NPP (Visible Infrared Imaging Radiometer Suite on the Suomi National Polar-orbiting Partnership Satellite) and Luojia1-01 can be obtained from 2012 and 2018, respectively. The new generation of NTL data, VIIRS-NPP and Luojia1-01 provide high-quality nighttime images with a spatial resolution of around 500 m and 130 m [65], respectively, which is higher than that of DMSP/OLS, and demonstrate significant advantages over DMSP/OLS imagery. Nowadays, several studies have been performed to combine NTL data from DMSP/OLS and NPP-VIIRS. Different integrating methods were adopted, such as the power function model [66], geographically weighted regression approach [67], sigmoid function model [68,69], and machine learning method [70]. Meanwhile, there are no unified criteria for NTL data from the two datasets simulated to DMSP-like or VIIRS-like NTL data. As a result, there is no mature and widely accepted method integrating the two NTL datasets yet. At the same time, the long-time series DMSP/OLS NTL data spanning from 1992 to 2013 is still a unique and valuable dataset with many researchers focusing on it currently [71,72]. Further researches are needed to investigate the most appropriate method combining DMSP/OLS and NPP-VIIRS NTL data. Otherwise, the emerging NTL data sources can be applied to monitor the latest urbanization separately, which is suggested for future studies.

## 6. Conclusions

Africa has experienced rapid urbanization over the last few decades and remains one of the fastest urbanization regions currently. This study presents an overall analysis of urbanization dynamics at the continent, sub-region, country, and local scales across Africa using DMSP/OLS NTL data from 1992 to 2013. After NTL calibration, we quantitatively partitioned the yearly DMSP/OLS imagery across Africa into three categories: low, medium, and high nighttime lighting areas based on the BG method. Lighted areas have grown enormously across Africa, especially low nighttime lighting areas. Five countries, including South Africa, Algeria, Nigeria, Egypt, and Libya have been leading the brightening trend of Africa, covering more than half of the total nighttime lighting areas in Africa both in 1992 and 2013. The spatiotemporal transitions of urban lighting are in concentric rings with high nighttime lighting areas in the urban center and low nighttime lighting areas in the urban fringe.

Results from this study show that nighttime light datasets are good sources to analyze and monitor urbanization, especially for those areas with few statistical materials, like Africa. Africa is in the process of rapid urbanization. However, there are some problems

along this, such as unsustainable land use and urban poverty. To make urbanization a potential stimulus to the socioeconomic and development of Africa, powerful and sustainable urbanization policies are urgently needed. Considering that Africa's urbanization level is estimated to rise, how to improve urbanization quality and sustainable development in Africa needs further study.

**Author Contributions:** S.J. and Z.Z. conceived and designed the research topic; S.J., G.W. and P.D. carried out the method and processed the data; S.J. prepared and wrote the original draft; Q.W., Y.W., M.X. and B.L. reviewed and edited the paper. All authors have read and agreed to the published version of the manuscript.

**Funding:** This research was funded by "Fundamental Research Funds for the Central Universities" (No. 14370401) and the project commissioned by the International Scientific and Technological Innovation Cooperation Key Project of the National Key R&D Program (No. 2018YFE0105900).

**Institutional Review Board Statement:** Not applicable.

**Informed Consent Statement:** Not applicable.

**Data Availability Statement:** Publicly available datasets were analyzed in this study. This data can be found here: https://www.ngdc.noaa.gov/eog/download.html

**Acknowledgments:** The authors express thanks to anonymous for their constructive comments and advice.

**Conflicts of Interest:** The authors declare no conflict of interest.

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
