# Peer review of "Detecting the Dynamics of Urban Growth in Africa Using DMSP/OLS Nighttime Light Data"

_land, doi:10.3390/land10010013_

Round 1

Reviewer 1 Report

Generally speaking, this work has an appropriate scientific level and I would judge its study objective to be appropriate and in line with the title and subject of the publication. I similarly think that one of the strong points of this proposal is its originality, both in terms of the subject treated and the hypotheses analysed.

This study covers a matter of current relevance

I appreciate the measured use of the bibliography cited and how this largely focuses on works that are relevant to the subject and well-contrasted. The structure used is in line with accepted practice and follows the prescribed order for a scientific article. Once the points outlined below have been corrected, this work should have an appropriate writing style and correct syntax and grammar. The main core of the work is based on Africa, which is the least urbanized region in the world, with only 43% of its population 351 living in the urban. At the same time, Africa has a relatively high urban population growth rate, and 352 there will be 1.5 billion urban dwellers in Africa by 2050, which means there will be a rapid 353 urbanization process from 2018 to 2050 in Africa.

Author Response

Authors would like to thank the Reviewer for his/her kind attention to the paper and constructive comments to improve the work. According to the suggestions of the reviewers, we have carefully checked our paper, and all the problems have been revised. We thank the reviewer for his/her helpful suggestion again. The sentences “Currently, Africa is the least urbanized region in the world, with only 43% of its population living in the urban. At the same time, Africa has a relatively high urban population growth rate, and there will be 1.5 billion urban dwellers in Africa by 2050 [34], which means there will be a rapid urbanization process from 2018 to 2050 in Africa.” have been replaced by “The main core of the work is based on Africa, which is the least urbanized region in the world, with only 43% of its population living in the urban. At the same time, Africa has a relatively high urban population growth rate, and there will be 1.5 billion urban dwellers in Africa by 2050[37], which means there will be a rapid urbanization process from 2018 to 2050 in Africa.” in line 380-383 in the revised manuscript as suggested by the reviewer.

Finally, we would once again like to thank Reviewer 1 for reviewing the paper and for his/her kind comments.

Reviewer 2 Report

The manuscript entitled “Detecting the dynamics of urban growth in Africa using DMSP/OLS nighttime light data” is a well-written paper that defines a new quantitative evaluation of urbanization trends and patterns while using night lightening as a measure of urban growth.

The study employes an analysis of a long time series sat acquisition, calibrated and observed from 1992 to 2013.

I think this argument is of high interest since the Land Use Change detection in Africa suffers from data paucity and difficulties in quantifying the exact amount n land take. Thus this study is of high interest.

I think that the structure is robust and the manuscript is easy-to-understand, which facilitate the comprehension of results. Event the methodology is quite simple since there are no obscure passages (my only doubt concern the utilization of NDVI to correct your index since you didn’t declare at which threshold NDVI has been used to clear your DN values).

I found the results a bit confused, meaning that you can create subchapters to present the results of your analysis better, while the discussion to me lack of a connection with other African studies on urban growth that proves if your quantities are matching other predicted or detected trends or not.

In the end, I think that working on these minor issues will improve quality significantly.

You can see my detailed comments in the attached file.

Good luck!

Author Response

Reviewer 2:

Comments to the Author:

The manuscript entitled “Detecting the dynamics of urban growth in Africa using DMSP/OLS nighttime light data” is a well-written paper that defines a new quantitative evaluation of urbanization trends and patterns while using night lightening as a measure of urban growth.

The study employes an analysis of a long time series sat acquisition, calibrated and observed from 1992 to 2013.

I think this argument is of high interest since the Land Use Change detection in Africa suffers from data paucity and difficulties in quantifying the exact amount n land take. Thus this study is of high interest.

Author’s Comments:

Authors would like to thank the Reviewer for his/her kind attention to the paper and constructive comments to improve the work. We have addressed all your comments in the modified version of the paper. Changes made in the modified version are highlighted to ease the review process. Also, we have added detailed response, in the following text, to each of your comments for better illustration. We believe that the revised manuscript is much better now after implementing all the reviewers’ suggestions.

R2.1: I think that the structure is robust and the manuscript is easy-to-understand, which facilitate the comprehension of results. Event the methodology is quite simple since there are no obscure passages (my only doubt concern the utilization of NDVI to correct your index since you didn’t declare at which threshold NDVI has been used to clear your DN values).

Response:

Thanks for pointing out the issue. The range of NDVI values have been addressed in Section 3.1.4. Meanwhile, we have added Figure 5 to show the effects of NDVI-induced calibration.

Please refer to Section 3.1.4 (line 177-179) and Section 4.1 (line 241) for the updated content and figure.

R2.2: I found the results a bit confused, meaning that you can create subchapters to present the results of your analysis better, while the discussion to me lack of a connection with other African studies on urban growth that proves if your quantities are matching other predicted or detected trends or not.

Response:

We thank the reviewer for the suggestions. We have created two subchapters “4.2.1 Continental and sub-regional level trends in different nighttime lighting areas” and “4.2.2 Leading countries of different nighttime lighting types” for Section 4.2 according to the reviewer’s suggestion. In 4.2.2, we showed the increased areas and average annual growth rate separately. As for the discussion, sustainable urban development is critical for Africa. Unsustainable land-use and urban poverty are two major concerns of urban sustainability in Africa. And our study can relate to these two issues, so we discussed them in our discussion. As suggested by the reviewer, we have added some content to compare our study with other researches to confirm our study. We thank the reviewer for his/her helpful suggestions again.

Please refer to Section 4.2 (line 247) and Section 5.2 (line 387) in the revised manuscript for the change.

In the end, I think that working on these minor issues will improve quality significantly.

You can see my detailed comments in the attached file.

Good luck!

Other comments in the attached file:

Comments in the formal manuscript

Revised manuscript

Re-sentence

“It has been found that the conversion of nighttime lighting types conformed to the characteristics of urban development with high nighttime lighting types showed up at the urban center, whereas medium and low nighttime lighting types appeared in the urban-rural transition zone and rural areas respectively.”

Replaced by: (line 23)

“The distribution of nighttime lighting types was consistent with the characteristics of urban development, with high nighttime lighting types showed up at the urban center, whereas medium and low nighttime lighting types appeared in the urban-rural transition zone and rural areas respectively.”

data is repeated 3 times:

“Statistical data, like socioeconomic and census data, are useful auxiliary data in urbanization studies”

Replaced by: (line 38)

“Statistical variables, such as socioeconomic and census data, are useful and auxiliary in urbanization studies”

see spacing:

“by various researchers[7,10,11]”

Has added a space: (line 43)

“by various researchers [7,10,11]”

is not the product that focuses on LULC, but the kind of utilization of this product is focused on LULC (for many reasons)

Has been corrected: (line 45-46)

The above satellite products are mainly applied to monitor the land-use and land-cover change in urban areas.

"It" refers to "the utilization of NTL? if so, please write down extensively

Reply:

“It” refers to “using long-term anthropogenic nighttime light satellite datasets”. We have added some content before and after this sentence. Please refer to line 49-60.

DN stay for?

DN stay for digital number, we have added it in the revised manuscript (line 70).

“means” to “implies”

Have been changed in line 104

Question about NDVI

Have replied above

“of about 2013”

Have been corrected in line 131

Lacks of

My grammar-check tool will report error when I added the of, so I didn’t add it in my paper.

it means that you have two scans of the indicator for the same year? If so, please be more clear

Reply: (line 154)

This information is available in Table 2. I add the information of it to inform my reader.

Question about NDVI

Have replied above

this is introductory, you should move this above when you first inroduce BG

Reply:

I find it's a repetition of line 68-76, so these sentences have been removed.

don't use obvious (since otherwise there is no need to mention). better "clear"

Reply:

Thanks for the suggestion. I have changed this sentence to “Figure 4a shows the total DN values of the original NTL data before calibration, from which we can see that the raw NTL data lacks continuity and comparability.” in line 219-220.

Confirm to confirming

Have been changed in line 222

Re-sentence:

As seen in Figure 5b-f, three different nighttime lighting types in five sub-regions were all in an increasing trend.

Reply:

Have been re-sentenced “The spatiotemporal trends of three different nighttime lighting types in five sub-regions (Northern, Western, Middle, Eastern, and Southern Africa) were presented in Figure 6b-f, respectively.” in line 261-263

increased increasing

This part has been major revised

i think you sould organize this discussion in subchapters:  one deals with the growht in macroregions north sud east and west. One tells the ligting index in the forst yar and the last year. One deals with the rate of variation. Each divided into your lightening categories...

Have replied above. Thank you for your helpful suggestions again.

can you also provide a map for certain data? this will hel your discussion

Figure 7 and Figure 8 are supplementary figures.

this is introductory and general (avoid here):

“Therefore, the urban development trajectory can be seen from the spatial evolution of artificial nighttime light”

This sentence has been removed. Thank you for your reminding.

Obvious-clear

This sentence has been re-sentenced “The arterial roads were mainly located in the areas with high nighttime lighting types.” In line 375

but you don't measured it in your research empirically...

Reply: (line 376-378)

I have re-sentenced this sentence by “Positive spatial correspondence can be visually inspected between high population density and high nighttime lighting types, which verified that DMSP/OLS NTL was related to intensified human activity.”

but rather than opening up to these broad issues why don't try to see if your quantitative results match with come other broad studies on African growth??? because you are introducing a new method that could be potentially used in the future...

Have replied above. Thank you for your valuable suggestions again.

Finally, we would once again like to thank Reviewer 2 for reviewing the paper and for his/her kind comments.

Reviewer 3 Report

The article entitled "Detecting the dynamics of urban growth in Africa using DMSP/OLS nighttime light data" reports results on assessing urban dynamics in Africa via satellite night-time imagery. The authors utilize a brightness gradient method, suggested in (Ma et al., 2015) to partition the study area into low, medium, and highly urbanized districts and to assess their 'transitions' from one category to another. Although the approach seems to be interesting, some principal questions arise.

First and foremost, the main conclusion on "built-up areas in Africa have increased rapidly, particularly those areas with low nighttime lighting types", reported also in Fig. 5, might have resulted from the quality of the DMSP-produced data. This may be the case when highly lit areas did not demonstrate high growth due to the saturation problem of the night-time lights data. The authors mention that they perform NDVI-induced calibration procedure to overcome the saturation problem, but it is difficult to agree that 8 km resolution NDVI data would help to solve the problem. Could the authors report descriptive statistics of DN4 (formula (4) at p. 5) and DN3 (formula (3) at p.5)?

Second, it would help a lot to understand the results better if the authors would plot BG versus DN levels (formula (8) at p. 6). It would be more convincing for the partition of the area actually explained by Fig. 3. Actually, this partition is the basis for all the conclusions and the reader should understand the principle better.

Third, as far as I understood, refer to urban dynamics in the cases when they actually mean the level of lights emitted from the areas. I did not find any statistically significant comparison with data presenting the ground truth. I would suggest the authors elaborate on the issue.

Finally, the limitation mentioned by the authors (Section 5.3) is extremely essential. Nowadays, there exist research aiming to combine DMSP and VIIRS-provided NTL data

(see Zhao, M., Zhou, Y., Li, X., Zhou, C., Cheng, W.. Li, M., Huang, K. Building a Series of Consistent Night466 Time Light Data (1992–2018) in Southeast Asia by Integrating DMSP-OLS and NPP-VIIRS. IEEE Transactions on Geoscience and Remote Sensing 2020, 58, 1843-1856;

Zheng, Q., Weng, Q., Wang, K. Developing a new cross-sensor calibration model for DMSP-OLS and Suomi-NPP VIIRS night-light imageries. Isprs Journal of Photogrammetry and Remote Sensing 2019, 153, 36-47).

I suggest the authors discuss it deeper.

I also have some non-crucial comments:

1) I am not sure that the spatial resolution of the DMSP product is 1 km (line 109, p.3). Please check.

2) Table 4 and the text before it (lines 287-302) is difficult to read.

3) Figs. 7 and 8 - it is difficult to consider it as an argument for any statement. Could the authors provide any statistics?

Round 2

Reviewer 3 Report

I appreciate the effort the authors made to improve the manuscript.